# Incidences for Fractures 2017–2021: What Do We Learn from the COVID-19 Pandemic?

**DOI:** 10.3390/healthcare11202804

**Published:** 2023-10-23

**Authors:** Ulrich Niemöller, Christian Tanislav, Karel Kostev

**Affiliations:** 1Department of Geriatrics and Neurology, Diakonie Hospital Jung Stilling Siegen, 57074 Siegen, Germany; ulrich.niemoeller@diakonie-sw.de; 2Epidemiology, IQVIA, 60549 Frankfurt am Main, Germany; 3Institute for Healthcare Research and Clinical Epidemiology, Philipps University Marburg, 35037 Marburg, Germany

**Keywords:** fracture, risk of fall, COVID-19 pandemic

## Abstract

Purpose/Introduction: In the present study, we aimed to assess the long-term incidence of fractures and during the COVID-19 pandemic. Methods: The current cohort study included patients who had received an initial fracture diagnosis of any type documented anonymously in the Disease Analyzer database (IQVIA) between 2017 and 2021 by physicians in 941 general practices in Germany. We investigated the development of fracture incidence over this period. Results: A total of 196,211 patients had a fracture diagnosis between 2017 and 2021. The number of patients with fracture diagnosis was highest in 2019 (n = 50,084) and lowest in 2020 (n = 46,227). The mean age of patients increased from 60.8 years in 2017 to 63.3 years in 2021. Between 58% and 60% of patients were female. From 2017 to 2019, the number of fractures documented in the younger age categories remained constant. Between 2019 and 2020, an incidence decrease was documented in the younger age groups (age group 16–40 years: −17.17%; age group 41–60 years: −18.71%; age group 61–80 years: −6.43%). By contrast, a slight increase of 3.03% was identified in the age group >80 years of age. No relevant changes in fracture incidences were noted between 2020 and 2021. Incidence rates decreased for both sexes from 2019 to 2020 (female patients: −6.27%; male patients: −10.18%). In the youngest age group (16–40 years), the decrease observed in 2020 was due to lower incidences for fractures of the upper and lower extremities (−11.9%; −12.5%) and ribs (−50.0%). In the age group ≥80 years, fracture incidences increased for the upper extremity (+2.8%), lower extremity (+8.3%), and femur (+8.3%). Conclusions: The circumstances of the pandemic reduced the incidence of fractures in younger people, probably due to reduced recreational activities, while fracture incidence increased in older people, presumably as a result of lack of support.

## 1. Introduction

Measures to combat the impact of the severe acute respiratory syndrome coronavirus 2 (SARS-CoV-2) pandemic, such as the reduction in seating in indoor spaces, social distancing, the mandatory utilization of masks, and “stay at home” campaigns, had dramatic consequences for social life worldwide [1,2]. The pandemic was observed to have affected various disorders in different ways, with similar trends emerging in many countries indicating that these effects were global in nature [3,4,5,6]. Incidence rates for various noninfectious acute medical conditions and disorders such as stroke and myocardial infarction decreased during the coronavirus disease 2019 (COVID-19) pandemic, while other non-coronavirus disease 19 (COVID-19) infectious diseases such as respiratory tract infections and gastro-intestinal infections also occurred less frequently due to the implementation of strict hygiene rules [3,4,5,6]. Pathologies such as myocardial infarction or stroke were less frequently documented in clinical databases and outpatient practices, probably because many mild ailments went untreated during the pandemic due to patients’ general reluctance to seek medical care [7]. By contrast, fractures resulting from a traumatic injury would most likely require an immediate medical consultation, which may explain why their occurrence might remain unaffected by the pandemic environment. Nevertheless, data relating to this assumption remain contradictory. While some authors reported an increase in fractures during the pandemic, others depicted a decrease [8,9,10,11].

For this reason, we aimed to investigate the incidence of fractures during the COVID-19 pandemic in the present study using outpatient data from a large database populated by general practitioners and pediatricians in Germany.

## 2. Methods

This retrospective cross-sectional study used data from the Disease Analyzer database (IQVIA), of which full details have been published elsewhere [12]. The Disease Analyzer database is composed of sociodemographic, diagnosis, and prescription data obtained in general and specialized practices in Germany. This database covers approximately 3000 private practices in Germany and has been shown to be representative of private practices in this country [12]. 

All individuals aged ≥16 with at least one visit to one of the 941 general practices (GP) across Germany represented in the database between January 2017 and December 2021 were included. Each practice delivered data continuously between 2017 and 2021. The study outcome was the number of patients with at least one fracture diagnosis (ICD-10: S02, S12, S22, S32, S42, S52, S62, S72, S82, S92, T02, T08, T10, T12) per practice in each year between 2017 and 2021, which enabled us to evaluate the related trends. These analyses were performed separately for four age groups (16–40, 41–60, 61–80, and >80 years) as well as separately for women and men. In addition, fracture sites categorized into upper extremity, lower extremity, spine, ribs, femur, and others were displayed separately for four age groups.

This study used descriptive statistics. Since the very large counts involved meant that every small difference would become significant, differences between the periods were not assessed using statistical tests. Analyses were carried out using SAS version 9.4 (SAS Institute Inc., Cary, NC, USA).

Based on German law, anonymous electronic medical data can be used for research, provided certain conditions are met. This legislation allows the use of these deidentified records without obtaining written informed consent from the patients and approval from a medical ethics committee.

## 3. Results

A total of 196,211 patients had a fracture diagnosis between 2017 and 2021. The number of patients with a fracture diagnosis was highest in 2019 (n = 50,084) and lowest in 2020 (n = 46,227). The mean age of patients increased from 60.8 years in 2017 to 63.3 years in 2021. Between 58% and 60% of patients were female (Table 1).

The highest level of incidental fractures occurred in the age group 61–80 years (range 2017–2021: 16.7–16.0 patients per practice). From 2017 to 2019, the incidence of fractures documented in the younger age categories remained constant, showing only a slight variability (age group 16–40 years: 9.7–9.9 patients per practice; age group 41–60 years: 12.4–13.9 patients per practice; age group 61–80 years: 16.7–17.1 patients per practice; age group >80 years: 11.7–13.2 patients per practice). From 2019 to 2020, a decrease in fracture incidence was documented in the younger age groups compared to the previous year (age group 16–40 years: 9.9–8.2 patients per practice (−17.17%); age group 41–60 years: 13.9–11.3 patients per practice (−18.71%); age group 61–80 years: 17.1–16.0 patients per practice (−6.43%)). By contrast, a slight increase of 3.03% was identified in the age group >80 years of age. Between 2020 and 2021, no changes in fracture incidence were noted for the age groups 16–40 years, 41–60 years, and >80 years of age, while a slight increase of 2.5% was observed in the age group 61–80 years. Results are summarized in Table 2 and Figure 1 and Figure 2. 

Similar trends were observed in both female and male patients; a decrease in incidence was recorded for both sexes from 2019 to 2020 (female patients: 31.9–29.9 patients per practice, −6.27%; male patients: 22.6–20.3 patients per practice, −10.18%). In general, fracture incidence was slightly higher in female patients during the investigated period (range from 2017 to 2021 in female patients: 29.9 to 31.9 patients per practice; range from 2017 to 2021 in male patients: 20.3 to 22.6 patients per practice). Results are summarized in Figure 3.

The incidence of fractures of the upper extremities was highest in the younger age groups, while femur fractures predominated in the older age groups. In the youngest age group (16–40 years), the decrease observed in 2020 was due to lower incidences of fractures of the upper and lower extremities and ribs (upper extremity 4.2–3.7 patients per practice, −11.9%; lower extremity 3.2–2.8 patients per practice, −12.5%; ribs: 1.2–0.6 patients per practice, −50.0%). In the age group 41–60 years, the decrease observed in 2020 was due to lower incidences of fractures of the upper and lower extremities and ribs (upper extremity 4.4–4.0 patients per practice, −9.1%; lower extremity 4.2–3.8 patients per practice, −9.5%; ribs: 2.6–1.9 patients per practice, −26.9%). In the age group 61–80 years, the decrease in fracture incidence occurred across all defined categories (upper extremity: −7.0%; lower extremity −2.9%; spine: −11.1%; ribs: −7.4%; femur: −8.0%; other: −10.5%). In the age group ≥80 years, fracture incidence increased in several defined fracture categories (upper extremity: +2.8%; lower extremity: +8.3%; spine: +0%; ribs: +5.6%; femur: +8.3%; other: +0%). Results are summarized in Figure 4 and Table 3. 

## 4. Discussion

In our study, the yearly incidence of fractures from 2017 to 2019 showed no significant change over time; at the most, a slight increase of 1–10% could be confirmed over this entire period. In 2020, the first year of the pandemic, a strong decrease in fractures incidence was recorded in both age groups ≤60 years (16–40 years: −17.17%, 41–60 years: −18.71%). The decrease observed in the age group 60–80 years was not very pronounced (−6.43%), while an increase in fracture incidence was actually confirmed in 2020 in the age group >80 years (+3.03%). A common trend for all age groups was observed for 2021, with incidence rates stagnating at the previous year’s (2020) level. The incidence of fractures was generally higher in women than in men; here, the similar trend for the first year of the pandemic could be observed (an incidence decrease of 6.27% in female patients and an incidence decrease of 10.18% in male patients). This difference could be due to the fact that women have osteoporosis much more often than men.

In the defined age categories 16–40 years and 41–60 years, the decrease in incidence during the first year of the pandemic was due mainly to decreasing incidences of fractures of the upper and lower extremities and ribs. In contrast, in the age category 61–80 years, a decreasing trend that was distributed equally across all defined fracture types was noted. By contrast, increasing incidences of fractures of the upper and lower extremities, ribs, and femur were detected in the first pandemic year (2020) in the age category >80 years. 

With the increasing number of COVID-19 patients hospitalized in 2020, a sharp drop in the number of patients seeking medical care for different medical conditions was recorded in several countries [13,14]. Acute care emergency departments reported a decreasing number of patients presenting with strokes, myocardial infarctions, and a number of other infectious diseases [15,16,17,18,19]. In view of these developments, the different effects of the pandemic environment became the subject of much discussion [20]. Prompted by stay-at-home campaigns and personal concerns about the risk of infection when entering a medical facility, many individuals might have avoided seeking medical care [7,21]. On the other hand, increased awareness of hygiene might have reduced the spread of other infectious diseases. By contrast, suffering a fracture might often be the result of an accident, frequently requiring urgent medical consultation. Therefore, reasons for a decline in the incidence of fractures during the pandemic might lie elsewhere than the above. However, since we have clearly identified the effect of the pandemic on fracture incidences, other explanations for this must be invoked.

Among the younger age groups (<60 years of age), we observed the strongest decline during the first year of the pandemic (2020). This development may have been due to restrictions implemented to avoid contact between people and to prevent the spread of the coronavirus. It could be speculated that people in this age group engage in sporting activities more often than older people. Consequently, a higher risk of injury could also be assumed here. As previously reported, nearly 40% of fracture cases in young adolescents result from sports and recreational activities [22]. The restrictions implemented during the pandemic might therefore have also curbed sports activities for many individuals in this age group, leading to a real decrease in fracture incidence. The fact that the reduction in this age category was based on fractures in locations that are typical for sports injuries (upper and lower extremities and ribs) is also consistent with this explanation [23,24]. 

The situation in the age category >60 years must be regarded differently. In this group of individuals, the higher risk of falling and general need for assistance and support with daily activities with increasing age must be taken into account when analyzing the development of fracture incidences during the COVID-19 pandemic [25,26]. According to our study results in the subgroup of patients aged 61–80 years, the incidence of fractures decreased by around 6% in 2020 because of the pandemic, which is clearly a smaller decrease than those observed in the younger age groups. Since the majority of the individuals in this age group (61–80 years) are still active but are already developing a certain risk of falling and are also gradually starting to engage less often in sporting activities, restrictions implemented during the pandemic obviously had less of an impact on preventing fractures within this cohort than it did in younger age groups. This is also supported by the distribution of the fracture localizations in this age category, with fewer of those kinds of fractures typically related to sport injuries, while fractures of the femur are more common, which is more consistent with a fall mechanism in older people [27]. We found the opposite development in the group of individuals aged >80 years, even detecting an increase in fractures during the first year of the pandemic. Measures implemented during the pandemic to promote social distancing are probably responsible for this observation. It could be speculated that many older people >80 years of age who needed support did not receive it to the required extent due to those measures taken during the pandemic. 

In line with previous investigations, we observed slightly higher yearly incidences of fractures in women than in men in our study [28]. However, the trend observed during the first year of the pandemic (2020) was obvious in both sexes. Furthermore, in accordance with the findings of previous analyses, fracture incidence rates detected in the second year of the pandemic duplicated values observed in 2020 across all age groups investigated [29,30].

Our study is subject to a number of limitations, most of which are due to the chosen study design and therefore cannot be avoided. First, no data are available on the fracture reasons (for example, car accidents or accidents in work or sport). Second, the diagnoses in our database were coded using ICD-10 codes and therefore the possibility that certain diagnoses were misclassified cannot be excluded; however, this should happen only rarely. Third, the database used does not contain information of diagnosis methods (for example, radiological methods). Fourth, the Disease Analyzer database does not include data on socioeconomic status (e.g., patients’ education and income) and lifestyle-related risk factors (e.g., smoking, alcohol consumption, and physical activity), so these could not be included in our study. Fifth, the database does not contain hospital data or data on mortality. Sixth, no data on healthcare costs are available to investigate the reduction in these costs due to the decrease in fractures. Nevertheless, the large sample size and the use of data from a large real-world patient sample are among the strengths of this study.

## 5. Conclusions

In the first year of the COVID-19 pandemic, incidence rates for fractures declined. A significant reduction was observed in the young age groups, (around −17%), while this effect weakened with increasing age, culminating in increasing fracture diagnoses in the age group >80 years. Trends during the second pandemic year (2021) approximated those observed during the first (2020), with no further decrease noted in this year. In young people, the pandemic environment reduced the incidence of fractures, probably due to reduced recreational activities and sports. In older people, the lack of required support with daily activities resulted in an increase in the number of people suffering fractures.

## Figures and Tables

**Figure 1 healthcare-11-02804-f001:**
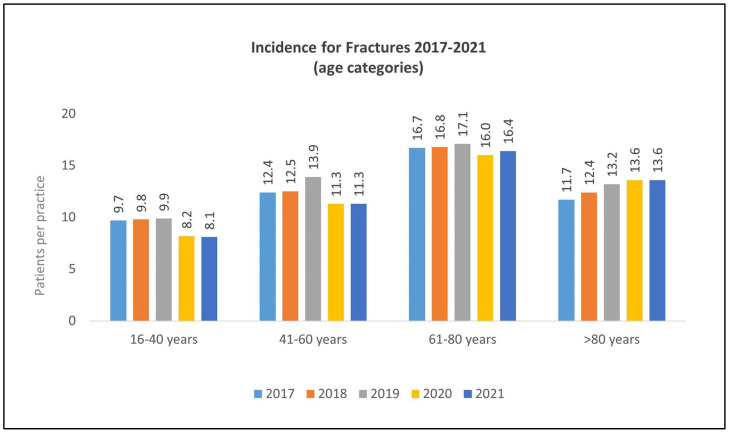
Fractures documented from 2017 to 2021, presented by age category.

**Figure 2 healthcare-11-02804-f002:**
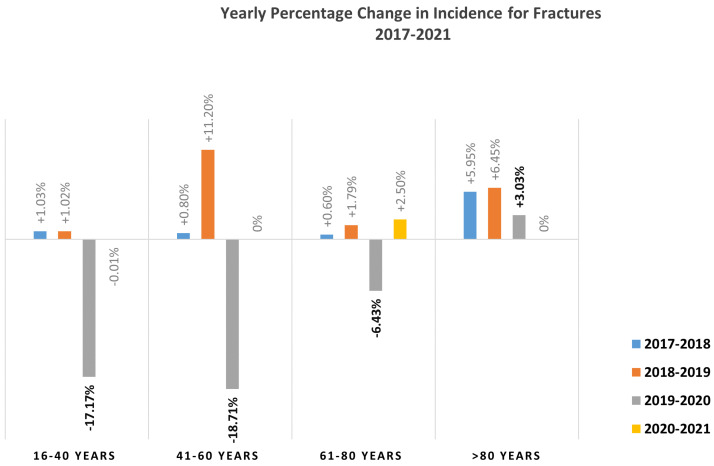
Change in fracture incidence from 2017 to 2021.

**Figure 3 healthcare-11-02804-f003:**
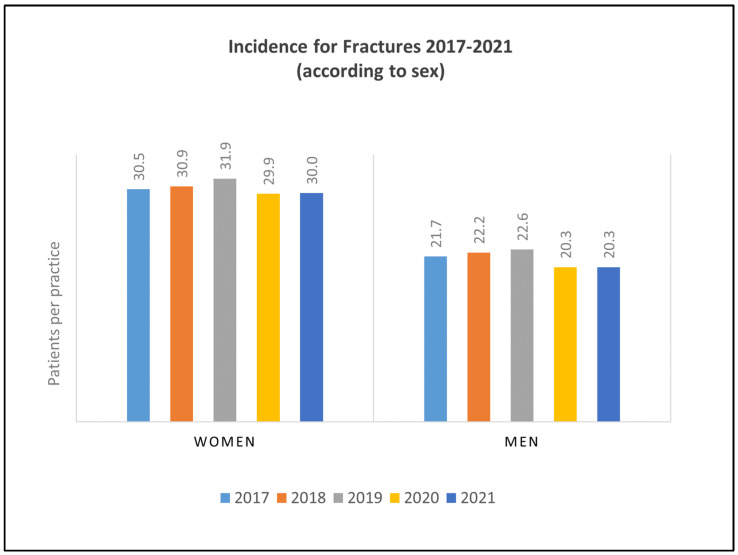
Fractures documented from 2017 to 2021, presented according to sex.

**Figure 4 healthcare-11-02804-f004:**
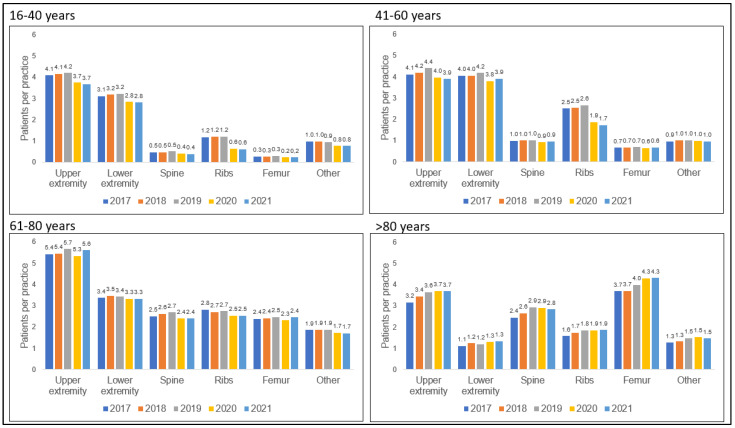
Fractures documented from 2017 to 2021, presented by age category and body region.

**Table 1 healthcare-11-02804-t001:** Age and sex characteristics of study patients.

Variable	2017	2018	2019	2020	2021
N	47,459	48,487	50,084	46,227	46,467
Mean age (standard deviation)	60.8 (16.2)	61.1 (16.1)	61.5 (15.9)	63.1 (14.6)	63.3 (14.9)
16–40 years (%)	19.2	19.0	18.6	16.7	16.3
41–60 years (%)	24.6	24.4	24.5	23.1	22.9
61–80 years (%)	33.1	32.6	32.1	32.6	33.2
>80 years (%)	23.2	24.0	24.8	27.7	27.5
Female (%)	58.4	58.2	58.6	59.5	59.7
Male (%)	41.6	41.8	41.4	40.5	40.3

**Table 2 healthcare-11-02804-t002:** Yearly absolute change in fracture prevalence (absolute values and percentages).

Fractures by Age Group		Yearly Difference			
		2018–2017	2019–2018	2020–2019	2021–2020
16–40 years	Patients per practice	9.8 − 9.7 = +0.1	9.9 − 9.8 = +0.1	8.2 − 9.9 = −1.7	8.1 − 8.2 = −0.1
	Percentage	+1.03%	+1.02%	−17.17%	−1.22%
41–60 years	Patients per practice	12.5 − 12.4 = +0.1	13.9 − 12.5 = +1.4	11.3 − 13.9 = −2.6	11.3 − 11.3 = 0
	Percentage	+0.80%	+11.2%	−18.71%	0%
61–80 years	Patients per practice	16.8 − 16.7 = +0.1	17.1 − 16.8 = +0.3	16.0 − 17.1 = −1.1	16.4 − 16.0 = +0.4
	Percentage	+0.60%	+1.79%	−6.43%	+2.5%
>80 years	Patients per practice	12.4 − 11.7 = +0.7	13.2 − 12.4 = +0.8	13.6 − 12.3 = +0.4	13.6 − 13.6 = 0
	Percentage	+5.98%	+6.45%	+3.03%	0%

**Table 3 healthcare-11-02804-t003:** Difference in fracture incidence rates during the first year of the pandemic (2020) in comparison to 2019 by fracture location.

	Percentage Difference for Incidences during the First Year of the Pandemic (2020) in Comparison to the Pre-Pandemic Year (2019)			
Fracture Location	16–40 Years	41–60 Years	61–80 Years	>80 Years
Upper extremity	−11.9%	−9.1%	−7.0%	+2.8%
Lower extremity	−12.5%	−9.5%	−2.9%	+8.3%
Spine	−20.0%	−10.0%	−11.1%	±0%
Ribs	−50.0%	−26.9%	−7.4%	+5.6%
Femur	−33.3%	−14.3%	−8.0%	+8.3%
Other	−11.1%	±0%	−10.5%	±0%

## Data Availability

The data that support the findings of this study are available on reasonable request from the corresponding author.

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
