# Peer review of "Incidences for Fractures 2017–2021: What Do We Learn from the COVID-19 Pandemic?"

_healthcare, 2023, doi:10.3390/healthcare11202804_

Round 1

Reviewer 1 Report

I would like to congratulate the authors for the impressive efforts on data collection and interpretation of the impressive number of patients. I do think the study will add important information on the epidemiology of the fractures during the pandemic.

Overall, there is a clear presentation of the data with simple statistics. However, I do believe that, the discussion might be more detailed in order to explain the findings.

1. Tab 3 is very important. Is there any more explanation why the frequency of the rib and femur fractures were very low up to 50%in the new ages group? Might be this related to less road traffic accidents and/or  less accidents in work! If there is available some statistics in Germany to justify this it will be clearly explained and pleasant for the readers. 

2. Regarding the spinal fractures in >80YO there was not change. Could be this related to the fact that, in this age the majority of the fractures are low energy , osteoporotic fractures , happening at home or care homes?

3. I am aware that, this study does not include costs and finance, but is there any information how much  the costs due to the lower incidence of the fractures might have decreased. Any comment regarding the costs might be important for the healthcare personal.

Author Response

I would like to congratulate the authors for the impressive efforts on data collection and interpretation of the impressive number of patients. I do think the study will add important information on the epidemiology of the fractures during the pandemic.

Response: Thank you very much for the positive feedback!

Overall, there is a clear presentation of the data with simple statistics. However, I do believe that, the discussion might be more detailed in order to explain the findings.

  1. Tab 3 is very important. Is there any more explanation why the frequency of the rib and femur fractures were very low up to 50%in the new ages group? Might be this related to less road traffic accidents and/or  less accidents in work! If there is available some statistics in Germany to justify this it will be clearly explained and pleasant for the readers. 

Response: We fully agree with the reviewer that the decrease of the rib and femur fractures could be traced back to less traffic accidents or accidents at work. Unfortunately, there is no citable statistic from Germany, and the publication (Yasin YJ, Grivna M, Abu-Zidan FM. Global impact of COVID-19 pandemic on road traffic collisions. World J Emerg Surg. 2021 Sep 28;16(1):51. doi: 10.1186/s13017-021-00395-8) gives some insights.  But as we do not have data on reasons for fractures, we would not like to speculate. However, we added the new limitation we probably forgot to mention last time “First, no data is available on the fractire reasons (for example, car accidences or accidences in work or sport).”

  1. Regarding the spinal fractures in >80YO there was not change. Could be this related to the fact that, in this age the majority of the fractures are low energy , osteoporotic fractures , happening at home or care homes?

Response: We think, no, as many patients in the database live in nursing homes, and as much we know from other publications, fractures are not rare in nursing homes.

  1. I am aware that, this study does not include costs and finance, but is there any information how much  the costs due to the lower incidence of the fractures might have decreased. Any comment regarding the costs might be important for the healthcare personal.

Response: Unfortunately, this only can be added as limitation, and we would like to investigate that in the future using other data.

Reviewer 2 Report

Thank you for the opportunity to review the manuscript: “Incidences for fractures 2017-2021: What do we learn from the COVID-19 pandemic?” by Niemöller et al. The manuscript has investigated incidences for fractures during the COVID-19 pandemic and adds important information on the subject.

Comments:

Line 13 Is anonymously the preferred word here, is it not rather deidentified data that is registered?

Line 14 and 62 Please clarify the number of general practices included as the stated 1,262 (line 14) or 941(line 62) are contradictory.

Line 54 Please clarify if the pediatricians are also in an out-patient setting.

Line 57 Please change the wording to the preferable “of which full details”.

Line 60 In what aspects has the IQVIA database been shown to be representative of the wider population?

Line 81 Please clarify how the diagnosis of the fractures registered had been made. By radiographs initiated by the GP or from a hospitalization or emergency room visit? Clinical diagnosis only?

Table 2 and figure 2 Please add information in table and figure legends why certain numbers are in bold.

Line 144 Even if incidence of fractures was higher in women than men, a larger decrease in incidence was seen for men during the pandemic, why would this be? Please add some comments to this observation.

Paragraph 166-176 In the IQVIA database is information on localization of the general practices available? Rural or urban areas? Could the changes in recreational habits be different depending on where the individual is living? Please add information on this if available.

Author Response

Reviewer 2

Line 13 Is anonymously the preferred word here, is it not rather deidentified data that is registered?

Response: We think, the most correct word is ‘anonymized’ what would describe the process. But finally, the data become “anonym” as a consequence of anonymization.

Line 14 and 62 Please clarify the number of general practices included as the stated 1,262 (line 14) or 941(line 62) are contradictory.

Response: That was corrected

Line 54 Please clarify if the pediatricians are also in an out-patient setting.

Response: That was clarified in text (outpatient)

Line 57 Please change the wording to the preferable “of which full details”.

Response: That was corrected

Line 60 In what aspects has the IQVIA database been shown to be representative of the wider population?

Response: We cited a reference of Rathmann et al. where there was a good agreement of the incidence or prevalence of major chronic diseases (diabetes, dementia, cancer) in the outpatient DA with German reference data.

Line 81 Please clarify how the diagnosis of the fractures registered had been made. By radiographs initiated by the GP or from a hospitalization or emergency room visit? Clinical diagnosis only?

Response: Database does not contain information of how diagnoses were made. We added that to Limitations.

Table 2 and figure 2 Please add information in table and figure legends why certain numbers are in bold.

Response: Bolding was removed.

Line 144 Even if incidence of fractures was higher in women than men, a larger decrease in incidence was seen for men during the pandemic, why would this be? Please add some comments to this observation.

Response: This difference could be due to the fact that women have osteoporosis much more often than men. That was added.

Paragraph 166-176 In the IQVIA database is information on localization of the general practices available? Rural or urban areas? Could the changes in recreational habits be different depending on where the individual is living? Please add information on this if available.

Response: Database does not contain this information, unfortunately.